# Functional Properties of Corn Byproduct-Based Emulsifier Prepared by Hydrothermal–Alkaline

**DOI:** 10.3390/molecules28020665

**Published:** 2023-01-09

**Authors:** Lu Liu, Jijun Zhang, Pengjie Wang, Yi Tong, Yi Li, Han Chen

**Affiliations:** 1Department of Nutrition and Health, China Agricultural University, Beijing 100083, China; 2National Engineering Research Center for Corn Deep Processing, Changchun 130000, China; 3Nutrition & Health Research Institute, China Oil and Foodstuffs Corporation, Beijing 102200, China; 4China Oil and Foodstuffs Biotechnology Corporation, Changchun 130000, China

**Keywords:** zein, mild hydrothermal–alkaline treatment, emulsifying properties, Turbiscan stability analysis, storage stability

## Abstract

As consumers’ interest in nature-sourced additives has increased, zein has been treated hydrothermally under alkaline conditions to prepare a nature-sourced emulsifier. The effects of mild hydrothermal–alkaline treatment with different temperatures or alkaline concentrations on the emulsifying properties of zein were investigated. The emulsification activity and stability index of zein hydrolysates increased by 39% and 164%, respectively. The optimal simple stabilized emulsion was uniform and stable against heat treatment up to 90 °C, sodium chloride up to 200 mmol/L, and pH values ranging from 6 to 9. Moreover, it presented excellent storage stability compared to commonly used food emulsifiers. The surface hydrophobicity caused the depolymerization of the tertiary structure of zein and the dissociation of subunits along with exposure of hydrophilic groups. The amino acid composition and circular dichroism results reveal that the treatment dissociated protein subunits and transformed α-helices into anti-parallel β-sheets and random coil. In conclusion, mild hydrothermal–alkaline treatment may well contribute to the extended functional properties of zein as a nature-sourced emulsifier.

## 1. Introduction

Proteins are natural surfactants customarily used to stabilize O/W emulsions, which can be adsorbed at the oil–water interface instinctively and dynamically [1]. As for the food industry, O/W emulsions are very common in branches of seasonings, sauces, batters, cheeses, creams, desserts, and dips. Corn is one of the most abundant agricultural commodities, and zein is the major protein in corn kernels and a byproduct of the corn industry, making it an accessible and versatile material [2]. Unlike other proteins, zein is highly hydrophobic and dissolvable in aqueous organic solutions [3]. Zein has been widely studied for stabilizing Pickering emulsions since there is a greater demand for biocompatible formulations for the pharmaceutical and food industries. To balance the extreme hydrophobicity of zein, incorporating other ingredients is needed, such as cellulose nanocrystals [4], polyphenols [5], and pectin [6,7]. However, such complexes are more stable in acid or alkaline environments and are easily influenced by salt concentrations and heating, commonly seen in food matrices, and the disassembly of the complexes [8,9]. The moderate alteration of its extreme hydrophobicity is vital to a required amphiphilic balance to fabricate a zein-based emulsifier [10].

Hydrothermal, a unique method traditionally used in geology or mineral crystal synthesis, uses high-temperature aqueous solutions at high vapor pressures. Recently, mild hydrothermal processes at normal pressure have drawn intense attention due to their safer and easier fabrication and simpler instrumental setups [11,12]. Previous studies indicated a positive correlation between the protein-stabilized emulsifiers’ thermal stability and functionality. Wang et al. [13] have found that alkaline and mild heating treatment can improve the emulsifying properties of hemp seed protein due to its subunit dissociation, tertiary-structure depolymerization, and hydrophobic-group exposure. However, the molecular weight (Mw) of hemp seed protein hydrolysates decreased from ~6.5–21 kDa to ~21–50 kDa, which was relatively high and led to a low emulsifying activity index of 7.39 ± 0.16 m^2^/g. Similar research was carried out on egg white protein having a lower emulsifying activity index of 3.05 m^2^/g [14]. There is hardly any research about the synergistic effect of alkaline and heating on the emulsification characteristics of zein.

On the other hand, it is widely documented that small hydrophobic peptides may result in a bitter taste [15]. The bitterness of peptides was predicted to increase before the Mw reached 1 kDa by quantitative structure–activity relationship models [16]. Similarly, several recent studies also suggested that the bitter taste of protein hydrolysates positively correlated with the peptide fraction between 0.5 and 3 kDa, especially in the 0.5–1 kDa range [17,18]. Therefore, screening polypeptides with a small Mw but larger than 3 kDa is vital for an emulsifier with high-emulsifying activity but no unpleasant taste. 

This study aimed to examine the synergistic effect of alkaline (0.05–1.0 M NaOH) and mild heating (55–80 °C)-induced transition of zein from a high hydrophobic state to a more hydrophilic state to produce a nature-sourced emulsifier with high emulsifying activity and stability. The measurement of emulsifying activity, emulsifying stability, Turbiscan Tower stability analysis, circular dichroism, surface hydrophobicity, particle size, and more importantly, the stability against environmental stress (pH, salt, and heating) were carried out to explore the relationship between structural changes and emulsifying properties, which is necessary when designing the proper zein-modifying protocols [13]. 

## 2. Results and Discussions

### 2.1. Tris-Tricine SDS-PAGE of Zein and Hydrolysates

The Mw distribution of zein and its hydrolysates prepared at different NaOH concentrations and temperatures were revealed by Tris-tricine SDS-PAGE (Appendix A). The bands at 19 and 22 kDa showed a decrease in intensity compared with that of zein, and broadband gradually appeared around 6.5 kDa with increasing processing strength, thus confirming the formation of polypeptides with lower Mw. When the NaOH concentration increased from 0.05 M to 0.4 M, the intensity of bands at 19 and 22 kDa decreased markedly, but they still existed until 0.6 M. The hydrolysis temperature showed a noticeable impact on Mw with a constant decrease from 55 to 80 °C with only a broadband between 3.3 and 6.5 kDa left at temperatures higher than 65 °C.

Yuan et al. [19] reported that heat treatment under pH 12.0 could effectively reduce the molecular weight of soy protein and the most susceptible temperature was 65 °C, similar to our results. Similar results were reported by Hui et al. [20] with proteins retrieved from waste sludge. On the contrary, heat treatment with mild alkali conditions would induce the aggregation of proteins. Mäkinen et al. [21] reported that disulfide-driven aggregations were not observed with extensive heating (100 °C) at lower pH (10.5). Hence, synergistic effects between alkaline and heating on the denaturation and destruction of protein subunits indicates that a controlled alkaline environment is critical to modifying zein molecules as required.

### 2.2. Surface Hydrophobicity Index (SHI) and Emulsifying Properties of ZHs

#### 2.2.1. Surface Hydrophobicity Index

Hydrophobicity is an essential factor affecting the three-dimensional structures of proteins [22]. Figure 1a,d presented the effects of thermal and alkaline treatment on the SHI of ZHs. The NaOH concentration and processing temperature showed a decreasing trend for SHI with increasing processing intensity. The SHI decreased from 7.55 × 10^4^ to 8.07 × 10^3^ with NaOH concentrations from 0.05 to 1.2 M and from 3.94 × 10^4^ to 8.75 × 10^3^ with temperatures from 55 °C to 80 °C, respectively.

When modified with alkaline, the tertiary structure of zein was looser, probably due to the damage to the hydrophobic interaction between zein molecules [23]. As a result, the hydrophilic groups buried among or inside the molecules were exposed, resulting in declining SHI. Treatment with NaOH concentrations higher than 0.6 M may further destroy the intramolecular interactions of zein due to its higher ability for deamidation, leading to the exposure of more hydrophilic regions inside zein and, therefore, the collapse of the tertiary structure. Rombouts et al. [24] concluded that alkaline treatment during the hydrothermal process of gliadin induced β-eliminations, a group of common side reactions which predominantly affect peptides with an electron-withdrawing substituent located on the sidechain, such as Cys or phosphorylated Ser/Thr. β-eliminations then deconstructed intra/inter-molecular disulfide bonds, resulting in the disassembly of protein molecules. 

Similarly, Thewissen et al. [25] reported that the increasing hydrophilicity of gliadin treated with an alkaline or acidic environment may contribute to the unfolding of protein molecules and result in the exposure of hydrophilic peptide chains. Consequently, repulsion among the molecules with exposed hydrophilic regions could indicate changes in the tertiary structure. On the other hand, ANS exhibits a stronger affinity to bind with pristine proteins than unorganized polypeptides [26]. Wang et al. [27] also proposed that intermolecular aggregation may mask hydrophobic sites and therefore decrease the surface hydrophobicity of proteins, which is unlikely in this study according to the results of SDS-PAGE. 

#### 2.2.2. Emulsification Activity Index and Emulsification Stability Index

The EAI and ESI of the ZHs increased significantly after alkaline hydrolysis (Figure 1). The EAI of ZHC and ZHT showed an overall increase trend (Figure 1b,e) and peaked at 0.6 M (85.41 m^2^/g) and 75 °C (88.54 m^2^/g), respectively, which was consistent with the results of SHI of hydrolysates. The ESI of ZHC and ZHT increased to 37.02 and 36.84 min, respectively, (Figure 1c,f) under the proper hydrolysis conditions (0.6 M NaOH and 70 °C). The emulsifying ability of the ZH improved with controlled hydrolysis due to the improvement of molecular flexibility, which corresponded to the deconstruction of zein molecules and low Mw [28]. The deconstruction of zein molecules resulted in protein unfolding, which destroyed the α-helix structure of the protein and increased its flexibility during the processing of heat-treatment-assisted alkali treatment [29].

The possible explanation for the decrease in EAI and ESI was that the peptides could aggregate due to the excessive hydrolysis that damaged the hydrophobic–hydrophilic balance [30]. The decrease in ESI was related to the decrease in viscoelasticity of the interfacial membrane caused by the fewer small peptides interacting at the interface, thus leading to the failure to maintain a stable interfacial membrane [26]. Additionally, the increased electrostatic repulsion due to excessive alkaline hydrolysis limited the structural expansion and reorientation of the low-Mw peptides at the interface. Consequently, proper alkaline hydrolysis improved the EAI and ESI, while excessive hydrolysis damaged the EAI and ESI of ZHs.

#### 2.2.3. Particle Size Distribution and Turbiscan Stability Index

The particle size distribution and back-scattered light map of emulsions stabilized by ZHCs and ZHTs are presented in Figure 2. With the increase in alkali concentration, the peak of particle size approached 1 µm; no secondary peak appeared when the concentration of NaOH was higher than 0.4 M. The stability of emulsions was also measured after storage at 25 °C for 24 h, and coalescence to different extents was observed. The coalescence decreased in the ZHC prepared with NaOH concentrations of 0.6 M. However, it increased with the secondary peak when the NaOH concentration was higher than 0.6 M. A similar trend was observed for ZHTs as particle size distribution showed no significant difference after preparation. Then, coalescence appeared after 24 h of storage at 25 °C, presenting a decreasing trend for ZHT prepared from 55 °C to 70 °C and increased when the hydrolysis temperature was higher than 70 °C. Proteins would gradually deconstruct during alkali–hydrothermal treatment, resulting in the collapse of most protein molecules. The increase in coalescence could be attributed to the excessive hydrolysis of zein, which exposed additional hydrophilic regions in the zein molecules, leading to the aggregation of the obtained hydrolysates into larger fragments by the action of chemical bonds during storage at room temperature [31].

The stability of emulsions stabilized by ZHs was examined using a Turbiscan Tower stability analyzer shown in Figure 2 as BSL maps along with the TSI at 24 h. Changes in the BSL profiles indicate sedimentation, coagulation, clarification, or creaming phenomena related to the unstable properties of goat milk samples during scanning; furthermore, the BSL profiles were calculated into the TSI values for direct comparisons [32]. The BSL profiles indicated that the primary destabilization mechanism in ZH-stabilized emulsions was flocculation, which had a similar pattern as reported by Huck-Iriart et al. [33]. The ZHC prepared with 0.6 M NaOH aqueous solution (Figure 2a) exhibited the least changes at the bottom (height at 0–10 mm), middle (height at 10–30 mm), and top (height at 30–42 mm) of the cylinder. The BSL in the 10–30 mm zone diminished by more than 2.5%, indicating changes in particle size in the system, which is concordant with the results for particle size. Similar trends were observed in Figure 2b, with ZHT prepared at 70 °C showing the least changes in the BSL profile, showing the lowest TSI value of 1.9. The alkali–hydrothermal process strongly affected the emulsion-stabilizing ability of ZHT, and the emulsion stabilized by ZHs prepared under excessive alkali–hydrothermal processing showed drastic destabilization [31].

### 2.3. Environmental-Stress Stability of Emulsions

#### 2.3.1. The Stability of the ZH_0.6–70_ Stabilized Emulsion against Ionic Strength

The continuous phase of food and emulsion systems contains many electrolytes (e.g., sodium, calcium, potassium, and phosphate), which may change the ionic strength of the emulsion system and thereby influence the stability of the emulsion [14]. To explore the ionic resistance of the emulsions, the ZH prepared using the optimum condition (0.6 M NaOH at 70 °C, ZH_0.6–70_) was employed to form an emulsion, as stated in Section 3.8. The particle size of the emulsions after 24 h of addition of NaCl can be found in Appendix A and Figure 3a. 

As shown in Appendix A and Figure 3a, the particle sizes of the emulsion droplets did not change with the increase in NaCl concentration below 50 mmol/L. D_[3,2]_ increased from 1.24 ± 0.01 μm to 1.40 ± 0.04 μm, and the specific surface area dropped from 4932.2 ± 8.38 to 4296.6 ± 15.08 m^2^/kg when the NaCl concentration increased from 50 to 100 mmol/L. Interestingly, the BSL profiles showed a small quantity of sedimentation and clarification with low content of NaCl, and the clarification decreased with an increase in sedimentation at a NaCl concentration of 500 mmol/L. The microscopic images (Figure 3b) were in accordance with the particle size results, showing inhomogeneous oil droplets at NaCl concentrations of 200 and 500 mmol/L. Therefore, the maximum tolerance of the emulsion to NaCl concentration was below 100 mmol/L. Wang et al. [34] studied the effect of ionic strength on the stability of enzymatically hydrolyzed rice bran albumin and globulin fractions. The emulsion particle size significantly increased with NaCl concentrations of more than 100 mM, similar to our study, which could be due to the strong salting-out effect at high salt concentrations. Electrostatic repulsion was not enough to overcome the attractive interactions between the droplets, which led to emulsion flocculation and increased particle size [35].

#### 2.3.2. The Stability of the ZH_0.6-70-_Stabilized Emulsion against Thermal Process

ZH_0.6–70_-stabilized emulsions were heated in a thermostatic water bath for 60 min at 70, 80, and 90 °C to explore the heat resistance of the emulsions. The particle size and BSL profiles characterized the stability of the emulsion at high temperatures. As shown in Appendix A and Figure 4a, the emulsions showed high stability to thermal treatments (up to 90 °C), and the particle sizes of the emulsions remained at ~1.50 μm, but the specific surface area decreased from 4893.40 ± 12.42 to 4026.20 ± 5.40 m^2^/kg after treatment at 90 °C with a statistical significance. The protein membrane adsorbed on the oil–water interface was destroyed under high-temperature conditions, so the stability of the emulsion decreased [36]. However, the emulsions of ZH_0.6–70_ had good thermal stability, and the particle sizes of the emulsions did not change significantly during the heating process. Sharma et al. [37] reported the hydrophilic–lipophilic balance formation and stabilization of emulsions. Thus, the amphiphilicity of ZHs provided a thermal protective layer for emulsions. The BSL profiles (Figure 4b) showed the same patterns before and after thermal treatments, indicating that thermal treatments had minor impacts on the stability of emulsions, which was in line with microscopic images (Figure 4b). The results confirm that ZH prepared by heating-assisted alkaline treatment is able to form emulsions with good thermal resistance.

#### 2.3.3. The Stability of the ZH_0.6–70_-Stabilized Emulsion in Different pH Environments

Appendix A and Figure 5 showed the effect of pH (3–9) on the stability of ZH_0.6-70-_stabilized emulsions. D_[3,2]_ of emulsions stabilized by ZH_0.6–70_ stayed at around 2 µm at a pH ranging from 6 to 9. D_[3,2]_ rapidly increased from 9.88 ± 1.13 to 138.2 ± 15.22 μm when the pH was adjusted from 4 to 5. The BSL profiles revealed changes in the uniformity of the emulsions within 24 h, as shown in Figure 5b. The BSL profiles of the emulsions prepared with pH 3 and 4 indicated sedimentation at the bottom (0–10 mm) and clarification at the top (30–42 mm). The low BSL could be due to unevenly dispersed particles existing in the emulsions. The stability of the emulsion improved at pH 5 with a small amount of sedimentation at the bottom (0–10 mm). The emulsions showed good stability when the pH was ≥6. The instability of emulsions at pH 4 and 5 could be explained by the decreased surface charge of proteins close to their isoelectric point (~4.5) [38]. This led to insufficient electrostatic repulsion for stabilizing the oil droplets, resulting in aggregation [39]. The microscopic images showed that the emulsions contained small droplets, uniformly distributed oil droplets in the pH range of 6–9 (Figure 5b), indicating that these systems were stable toward aggregation due to the strong electrostatic repulsion.

### 2.4. Evaluation of Storage Stability of the ZH_0.6–70_-Stabilized Emulsion

The storage stability of different emulsions was characterized by the changes in particle size after storing at 25 °C for 29 days (Appendix A). The mean particle size of all emulsions increased significantly after 29 days of storage (*p* < 0.05), presenting flocculation or coalescence within the matrices [40]. In this study, most emulsions’ volumetric mean particle sizes presented no significant changes after 15 days of storage, except for the one stabilized by E322, which showed a 159.8% enlargement. The ZH_0.6–70_-stabilized emulsion had the smallest mean particle size before (1.33 ± 0.02 µm) and after 29 days of storage (13.34 ± 3.73 µm) which indicated the storage stability at room temperature and was comparable to that of NaCN and E473. The better storage stability of the ZH_0.6–70_-stabilized emulsion could be due to its smallest particle size [41]. The particle size of the emulsion was determined by two main factors: homogenization conditions and the ability of the droplets to resist flocculation and aggregation [42]. Furthermore, this phenomenon could be explained based on ZH’s ease in forming a dense film around the droplets to prevent coalescence as compared to the three other emulsifiers [43].

### 2.5. Amino Acids Analysis

An overview of the AA composition of zein and ZH_0.6–70_ was presented in Table 1. All data are expressed in the percentage of cumulative AAs. We observed that glutamic acid was the most abundant AA in zein (24.19%) and ZH_0.6–70_ (24.69%), followed by leucine (19.20 vs. 18.61%), proline (9.2%5 vs. 8.32%), alanine (8.86% vs. 10.14%), phenylalanine (6.64% vs. 7.32%), and aspartic acid (5.11% vs. 6.18%). These AAs accounted for 73.25% and 75.26% of the total AAs in zein and ZH_0.6–70_, respectively. In addition, ZH_0.6–70_ exhibited significantly higher values than zein in alanine (8.86% vs. 10.14%), arginine (1.67% vs. 2.18%), aspartic acid (5.11% vs. 6.18%), histidine (1.59% vs. 2.33%), isoleucine (3.91% vs. 4.97%), and valine (3.28% vs. 4.73%) but lower content in serine (5.36% vs. 3.30%), threonine (2.97% vs. 1.22%), and methionine (1.50 vs. n.d.). The AA profile of zein agrees with the recent study [3]. The differences between AA profiles of zein and ZH_0.6–70_ were due to the fact that only the sediments were collected; therefore, some free AAs generated during hydrolysis were removed from the final samples [44,45]. The total hydrophilic AA content was 32.63% vs. 35.54% for zein and ZH_0.6–70_, respectively. The increase in hydrophilic AA content after proper treatment is in line with the decreasing surface hydrophobicity index and indicates the enhanced amphiphilic balance of zein hydrolysates.

### 2.6. Circular Dichroism

The spatial structure determines the functional properties of the protein. To explore the relationship between emulsifying properties and the secondary structure of ZHs, the effect of mild heating-assisted alkaline treatment on the secondary structure of zein and ZH_0.6–70_ was studied using circular dichroism. As shown in Table 2, the treatment changed the secondary structure of zein with decreased α-helix (from 51.13% to 14.02%) and increased anti-parallel β-sheet (from 1.75% to 35.11%) and random coil (from 27.26% to 33.08%) which is in accordance with the study of Usoltsev et al. [46]. Yu et al. [14] modified egg white protein with thermal alkaline treatment and noticed the potential disruption of hydrogen bonds and, consequently, the conversion from α-helix to β-turn. The α-helix is mainly responsible for maintaining the natural structure of proteins, and decreased α-helix indicates the denaturation and unfolding of the protein molecules [47]. A previous study reported that the soybean protein isolates with reduced α-helix content and increased β-sheet content presented higher hydrophilicity, which agrees with our results [48]. Therefore, the improved emulsifying properties of ZH_0.6–70_ could be explained mainly by the transformation of α-helix into an anti-parallel β-sheet, which led to changes in the secondary structure of zein [49].

## 3. Materials and Methods

### 3.1. Materials

Zein (CAS no. 9010-66-6) was purchased from Sigma-Aldrich LLC (St. Louis, MO, USA). Medium-chain triglyceride (MCT) oil was purchased from Jarrow Formulas Co. (Los Angeles, CA, USA). Pre-stained protein marker (3.3–31.0 kDa); Tris-tricine-SDS-PAGE loading buffer, 2 × (with DTT); Tris-tricine-SDS-PAGE gel-making kit; Tris-tricine-SDS-PAGE electrode buffer, 10 × Cathode buffer; Tris-tricine-SDS-PAGE electrode buffer, 10 × Anode buffer; and Coomassie brilliant blue R-250 were purchased from Solarbio Co. (Beijing, China). Sodium 8-anilino-1-naphthalenesulfonate (ANS-Na) was purchased from Aladdin Co. (Shanghai, China). Other chemicals used in this study were of analytical grade unless noted otherwise.

### 3.2. Preparation of Alkaline Hydrolyzed Zein Peptides

Zein hydrolysates were prepared according to two variables, namely temperature and NaOH concentration. Aqueous zein suspension (1% wt) was hydrolyzed with either various alkaline concentrations (ZHC, 0.2, 0.4, 0.6, 0.8, and 1.0 mol/L NaOH at 70 °C for 120 min) or different temperatures (ZHT, 55, 60, 65, 70, 75, and 80 °C with 0.6 mol/L NaOH for 120 min) in a magnetic stirring water bath with stirring speed set to 200 rpm. The samples were then cooled to room temperature using an ice water bath. The pH value was adjusted to 4.5 using 1 M HCl before the sediments were collected via centrifugation at 3984× *g*. for 10 min. The collected sediments were lyophilized for 48 h at −60 °C for further tests.

### 3.3. Tris-Tricine SDS-PAGE

The Tris-tricine SDS-PAGE was performed according to the method of Schägger. [50] Tris-tricine SDS-PAGE gels were made and run using a PowerPac Basic gel electrophoresis unit (Bio-rad, Hercules, CA, USA). The thickness of the gels was set to 1.5 mm, and each gel consisted of a ratio of 16.5% separating gel, 10% spacer gel, and 4% stacking gel at 6:1.5:2. Gels were run at 30 V until the tracking dye passed the stacking gel, and then a constant voltage of 100 V was applied until the dye reached the bottom of the gel. Gels were visualized by staining with Coomassie brilliant blue, and images of properly stained gels were captured by a gel imager (AI600 GEL Imaging System, GE Healthcare Co., New York, NY, USA). The gels were prepared according to the formulation shown in Appendix A.

### 3.4. Emulsification Activity Index (EAI) and Emulsification Stability Index (ESI)

The emulsifying properties of samples were determined following the turbidimetric method of Li et al. [51] with slight modifications. ZHs were dispersed in 10 mmol/L PBS (pH = 7.0), and then 15 g of solutions was mixed with 5 g of MCTs. The mixtures were emulsified for 1 min at 25,000 rpm by a high-speed dispenser (Ultra-Turrax T18, IKA, Staufen, Germany). Subsequently, 50 μL of sample emulsion was aspirated from the bottom of a plastic tube at 0 and 10 min and then mixed with SDS (5 mL, 0.10%, *w/v*) immediately. Finally, the absorbance of the samples was measured at 500 nm, with the SDS solution used as a blank (0.10%, *w/v*). The EAI (m^2^/g) and ESI (%) of the protein samples were evaluated as the following equations:(1)$${\mathrm{ESI}\;(m}^{2}/g)=\frac{2\times T\times A_{0}\times \mathrm{DF}}{10\times c\times \varphi }$$
where T represents the turbidity (2.303), A_0_ represents the absorbance at 0 min, DF represents the dilution factor (100), c represents the concentration of the protein solution before emulsification (mg/mL), and φ represents the oil-phase volume fraction (*v/v*) of the emulsion (0.25).
(2)$$\mathrm{ESI}\;(\min)=\frac{A_{0}}{A_{0}{-A}_{10}}\times 10$$
where A_0_ represents the absorbance at 0 min, and A_10_ represents the absorbance at 10 min.

### 3.5. Surface Hydrophobicity Index (SHI)

The SHI of ZH was estimated according to Song et al. [52], using ANS-Na as a hydrophobic fluorescent marker. The ZHs (0.05, 0.1, 0.2, 0.4, 0.6, 0.8, and 1.0 mg/mL) and ANS-Na (10 mmol/L) were dissolved in 10 mmol/L PBS (pH = 7.0). One milliliter of ZH solution was mixed with 10 µL of ANS-Na solution and vortexed for 30 s, and then the mixtures were incubated for 5 min at room temperature before measurement by a microplate reader (Synergy 2, BioTec Co., Beijing, China) using 360 and 460 nm as λ_ex_ and λ_em_, respectively, at 25 °C. The slope at the origin of the plot for fluorescence intensity versus protein concentration was used as the surface hydrophobicity index (SHI) [53].

### 3.6. Preparation of Emulsions

The prepared ZHs were dissolved in 10 mmol/L PBS (pH = 7.0) at a concentration of 0.3% (*w*/*w*), and the solution was then mixed with MCT at a solution/MCT ratio of 4:1 and pre-emulsified at 15,000 rpm for 1 min, using a high-speed dispenser (Ultra-Turrax T18; IKA, Staufen, Germany). The pre-emulsified mixture was homogenized under the pressure of 10 MPa three times using a high-pressure homogenizer (AH-BASIC 30; ATS Engineering Ltd., Suzhou, China) to obtain the final emulsion.

### 3.7. Stability of Emulsion

The stability of the emulsions was analyzed using a Turbiscan Tower stability analyzer (Formulaction Co., Toulouse, France) by scanning the emulsions from bottom to top using near-infrared light (λ = 880 nm) to determine the intensities of back-scattered light (BSL) and transmitted light. The Turbiscan Stability Index (TSI) is a specific parameter developed by Turbiscan Co. to compare and characterize the physical stability of various formulations with a single, comparable, and replicable number. Higher TSI values indicate lower sample stability during Turbiscan scanning [53]. The emulsions were left to rest for 5 min before testing. The scanning was conducted every 20 min for 24 h at 40 °C.

The was calculated according to the following expression [54]:(3)$$\mathrm{TSI}=\underset{t=1}{\sum }\frac{\sum_{h=0}^{h=H}\left|{\mathrm{BS}}_{t}\left(h\right)-{\mathrm{BS}}_{t-1}\left(h\right)\right|}{H}$$
where BS is the back-scatter intensity, t is the scanning time, h is the height of the measurement per 40 µm, and H is the height of the sample in the measuring cell. TSI and figures of BS were acquired via Formulaction Smart Science analysis software.

### 3.8. Particle Size

The particle size of emulsions was determined on a laser particle analyzer (Mastersizer 3000; Malvern Instruments, Malvern, UK). The samples were diluted 100 times with 10 mmol/L PBS to avoid the multiple-scattering effect caused by excess concentration. The refractive indexes of MCT and water were taken as 1.460 and 1.330, respectively [54].

### 3.9. Microscopic Observation

After preparation and 24 h storage, emulsions were observed under a bright field using a fluorescence microscope (DFC450C; Leica, Wetzlar, Germany). The emulsions were diluted 10 times using 10 mmol/L PBS before observation.

### 3.10. Emulsion Stability against Environmental Stress

The pH, thermal, and ionic strength stabilities of the freshly prepared emulsions were analyzed. The pH of the freshly prepared emulsions (pH 7) was adjusted from 2 to 9 by adding 1 M HCl or NaOH with continuous stirring. The sample was transferred into individual glass test tubes and incubated in a thermostatic water bath for 30 min at 70, 80, and 90 °C to investigate the stability of emulsions against the thermal process. The heated samples were cooled down immediately by placing them in ice water. Fresh emulsions were adjusted to ionic strengths of 10, 50, 100, 200, and 500 mmol/L by adding NaCl. The particle size and BSL profiles of treated samples were determined. The emulsions were photographed after 24 h storage at room temperature [26].

### 3.11. Emulsion Stability during Storage at Room Temperature

To simulate the practical use of emulsions, the changes in particle size of the ZH_0.6–70_-stabilized emulsion was compared to emulsions stabilized by three emulsifiers commonly used in the food industry (sodium caseinate, NaCN; soybean lecithin (E322); sucrose esters of fatty acid (E473)) during 29 days of storage at 25 °C [41]. The emulsions were prepared according to Section 2.6 and evaluated after preparation on day 15 and day 29.

### 3.12. Quantification of Total Amino Acids (AAs)

The composition of 17 Aas was determined based on Chen et al. [3] using an automatic AA analyzer (LA 8080, Hitachi, Japan) equipped with a high-performance cation exchange column (4.6 mm × 60 mm) after being hydrolyzed by 6 mol L^−1^ HCl with nitrogen. The wavelengths used were 570 nm and 440 nm. Sulfur-containing Aas were measured after performic acid oxidation. Identification and quantification of Aas were achieved by comparing the retention times of the peaks with those of respective standards. The aspartic acid and glutamic acid contents include asparagine and glutamine, respectively. For comparison, relative ratios of AA were calculated by normalizing against the total quantity of Aas.

### 3.13. Circular Dichroism

The change in the secondary structure of ZHs was measured using an MOS-500 CD spectrometer (Bio-Logic, Seyssinet-Pariset, France) under a constant nitrogen atmosphere in a quartz cuvette with a 0.1 cm path length over a scanning range of 180–260 nm. The step resolution was 1 nm, and the scanning speed was 60/min. The value of the 10 mmol/L PBS solution was subtracted from all CD spectral data. The change in the secondary structure of ZHs was calculated using CdtoolX (version 2.01) secondary structure estimation software [55].

### 3.14. Statistical Analysis

Unless specified otherwise, all experiments were conducted in triplicates. Data were analyzed using the general linear model and one-way ANOVA procedures of Minitab 19.0 (Analytical Software, State College, PA, USA). Figures that identified the significant differences (*p* < 0.05) between individual means were fabricated by OriginPro 2019b.

## 4. Conclusions

This study successfully fabricated a nature-sourced emulsifier based on a corn byproduct by mild hydrothermal–alkaline treatment. The optimum preparation condition was 0.6 mol/L NaOH at 70 °C. The Mw of the optimal ZH was distributed between 3.3 and 6.5 kDa. Its stability against different pH, ionic strength, and heating was evaluated. The prepared emulsions were uniform and stable against heat treatment up to 90 °C, sodium chloride up to 200 mmol/L, and pH values ranging from 6 to 9. The ZH_0.6–70_-stabilized emulsion exhibited excellent storage stability and was comparable to emulsions stabilized by NaCN and E473 with smaller particle sizes.

The alkaline–hydrothermal process profoundly changed the emulsification activity and stability of ZHs by dissociating protein subunits. Moreover, this process induced changes in the secondary structures of zein with decreased α-helix and increased anti-parallel β-sheet and random coil. The decrease in surface hydrophobicity index mainly reflected changes in the tertiary structure. Meanwhile, the peptides with bitter tastes are concentrated in the Mw range of 0.5–3 kDa, which is against our results. The high concentrations of alkaline were applied to explore the synergistic effects of the alkaline environment and thermal treatment. Further studies should focus on the milder fabrication conditions and controllable structure of zein hydrolysates with tolerance to broader pH values. In conclusion, mild heating-assisted alkaline treatment is a feasible method for adjusting the functional properties of zein, thereby expanding its use in the food industry as a natural emulsifier with no perceptible bitterness.

## Figures and Tables

**Figure 1 molecules-28-00665-f001:**
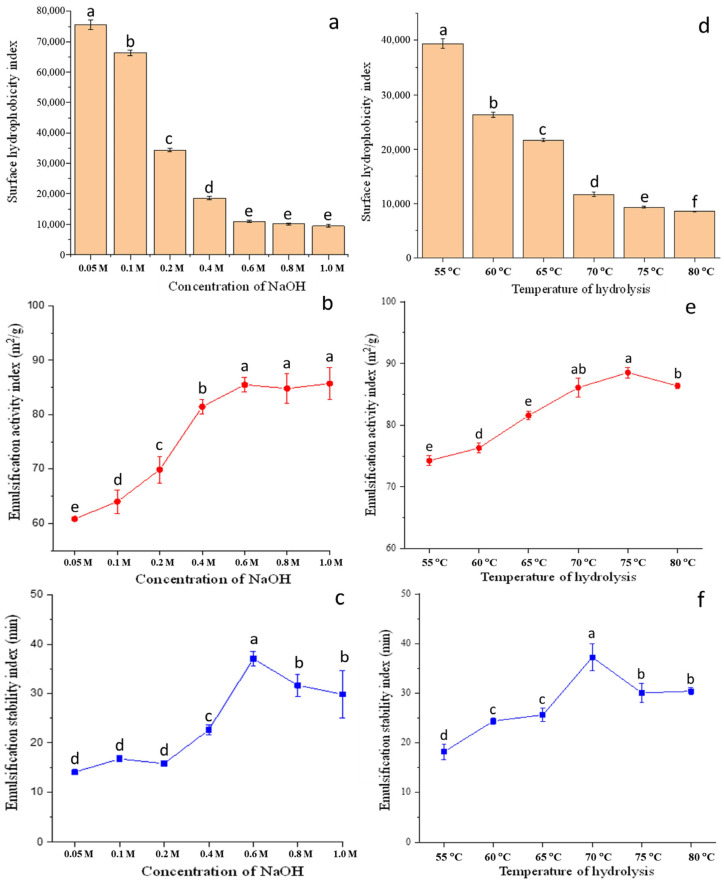
Surface hydrophobicity (**a**), emulsification activity index, (**b**) and emulsification stability index (**c**) of ZHCs and surface hydrophobicity (**d**), emulsification activity index, (**e**) and emulsification stability index (**f**) of ZHTs (n = 3). Different lowercase letters indicate a significant difference (*p* < 0.05).

**Figure 2 molecules-28-00665-f002:**
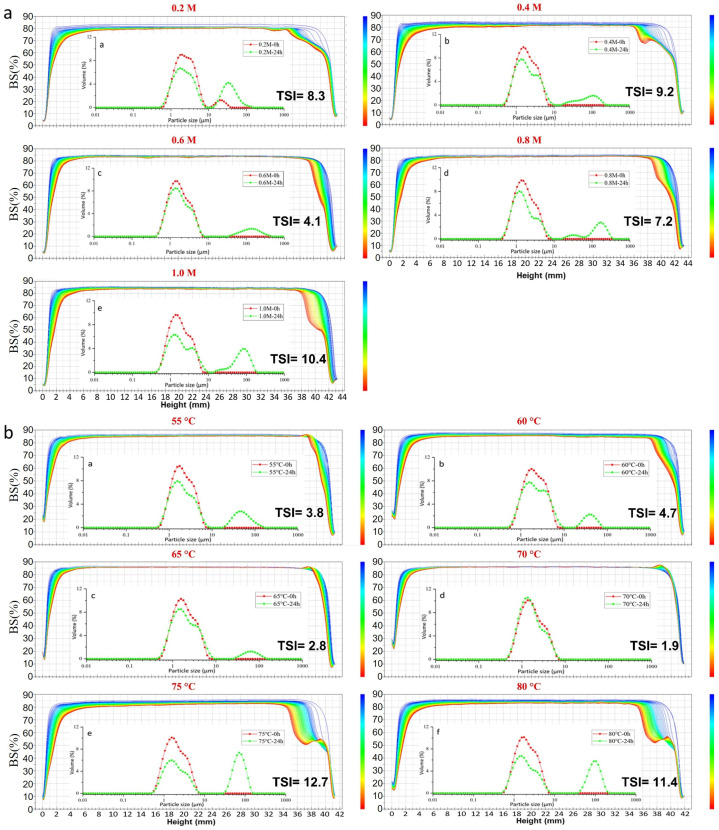
Back-scattered light map and size distribution of emulsions stabilized by ZHCs (a–e in (**a**) represent samples treated with 0.2 M, 0.4 M, 0.6 M, 0.8 M and 1.0 M NaOH at 70 °C, respectively) and ZHTs (a–f in (**b**) represent samples treated with 0.6 M NaOH at 55 °C, 60 °C, 65 °C, 70 °C, 75°C and 80 °C, respectively).

**Figure 3 molecules-28-00665-f003:**
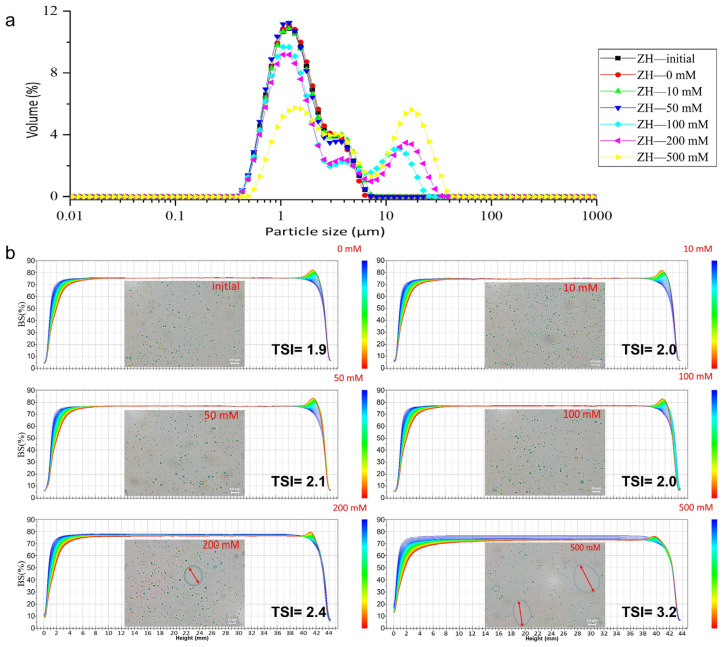
Particle size distribution of emulsions stabilized by ZH_0.6–70_ after adding different concentrations of NaCl (**a**); Back-scattered light map and microscopic images (scale bar = 10 µm) of emulsions stabilized by ZH_0.6–70_ after adding different concentrations of NaCl (**b**).

**Figure 4 molecules-28-00665-f004:**
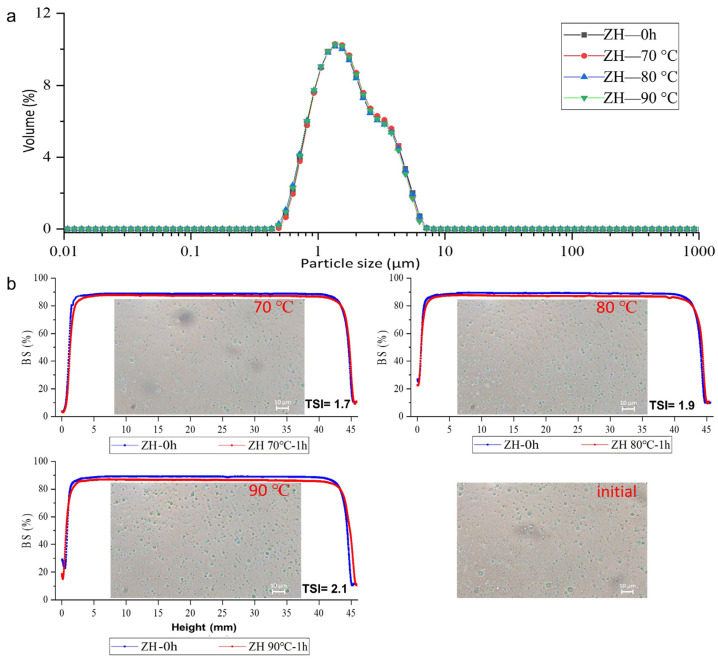
Particle size distribution of emulsions stabilized by ZH_0.6–70_ after thermal treatment (**a**); Back-scattered light map and microscopic images (scale bar =10 µm) of emulsions stabilized by ZH_0.6–70_ before and after heating under different temperatures (**b**).

**Figure 5 molecules-28-00665-f005:**
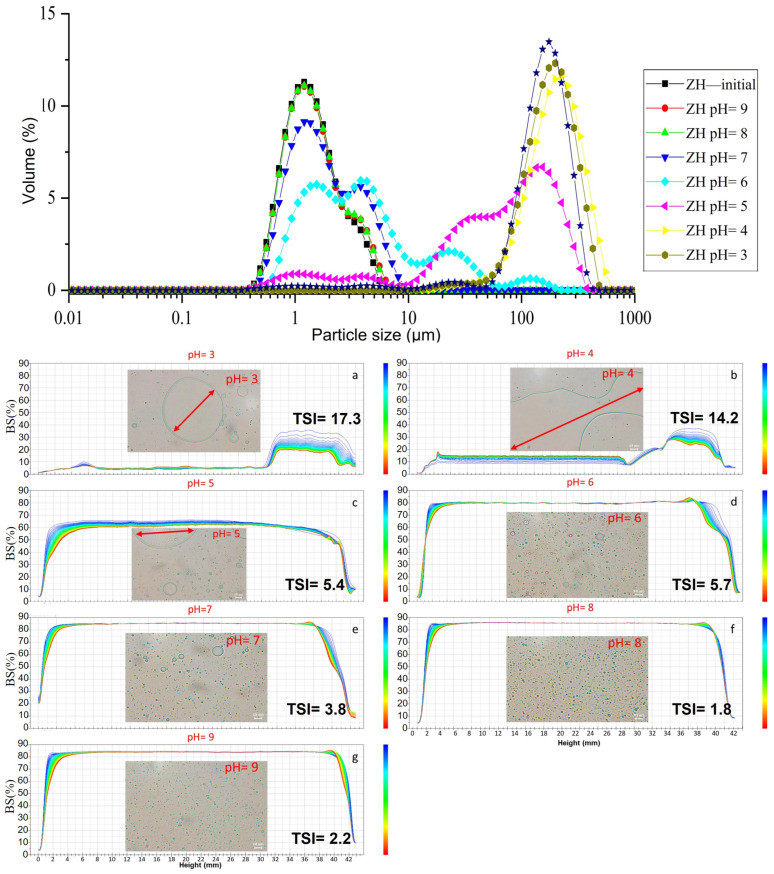
Particle size distribution of emulsions stabilized by ZH_0.6–70_ at different pH (**a**); Back-scattered light map and microscopic images (scale bar = 10 µm) of emulsions stabilized by ZH_0.6–70_ at different pH (**b**).

**Table 1 molecules-28-00665-t001:** Amino acids profiles of zein and ZH_0.6–70_ (n = 3).

Amino Acids	Zein	ZH_0.6–70_
Alanine	8.86 ± 0.22% ^b^	10.14 ± 0.27% ^a^
Arginine	1.67 ± 0.08% ^b^	2.18 ± 0.13% ^a^
Aspartic acid	5.11 ± 0.17% ^b^	6.18 ± 0.14% ^a^
Cysteine	0.54 ± 0.09% ^a^	0.06 ± 0.05% ^a^
Glutamic acid	24.19 ± 0.43% ^a^	24.69 ± 0.56% ^a^
Glycine	1.45 ± 0.21% ^a^	1.64 ± 0.14% ^a^
Histidine	1.59 ± 0.22% ^b^	2.33 ± 0.15% ^a^
Isoleucine	3.91 ± 0.16% ^b^	4.97 ± 0.34% ^a^
Leucine	19.20 ± 0.67% ^a^	18.61 ± 0.82% ^a^
Lysine	0.07 ± 0.04% ^a^	0.16 ± 0.07% ^a^
Methionine	1.50 ± 0.06%	n.d.
Phenylalanine	6.64 ± 0.55% ^a^	7.32 ± 0.44% ^a^
Proline	9.25 ± 0.42% ^a^	8.32 ± 0.54% ^a^
Serine	5.36 ± 0.22% ^a^	3.30 ± 0.13% ^b^
Threonine	2.97 ± 0.14% ^a^	1.22 ± 0.08% ^b^
Tyrosine	4.42 ± 0.13% ^a^	4.16 ± 0.24% ^a^
Valine	3.28 ± 0.41% ^b^	4.73 ± 0.27% ^a^
Total hydrophilic AA	32.63 ± 0.68% ^b^	35.54 ± 0.54% ^a^

n.d.: Not detected. Different superscript letters in the same row indicate a significant difference (*p* < 0.05).

**Table 2 molecules-28-00665-t002:** Secondary structure of four kinds of zein and ZH_0.6–70_.

	Random Coil	α-Helix	β-Sheet		β-Turn
			Parallel β-Sheet	Anti-Parallel β-Sheet	
Zein	27.26%	51.13%	6.28%	1.75%	13.58%
ZH_0.6–70_	33.08%	14.02%	3.00%	35.11%	14.80%

## Data Availability

Not applicable.

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
