# Peer review of "Functional Properties of Corn Byproduct-Based Emulsifier Prepared by Hydrothermal–Alkaline"

_molecules, 2023, doi:10.3390/molecules28020665_

Round 1
Reviewer 1 Report
The authors prepared a nature-sourced emulsifier based on a byproduct of corn industry and examined its emulsifying properties. It’s interesting to see a new way to explore the value of the agricultural byproducts. Please find the following questions need to be addressed by the authors:
1. Page 1 line 44, it seems that a reference was inserted twice by mistake as “[8,8,9]”, and some subscripts were mis-typed (e.g. page 7 line 229 and 230, D[3,2]), please fix mistyping issue through the manuscript.
2. Page 2 line 81-82, the sentence “With the increase in NaOH concentration from 0.05 M to 0.4 M, the intensity of bands that corresponded decreased markedly, but it still existed until 0.6 M.” is a bit of confusing, please rewrite the sentence to increase its readability.
3. Page 5 line 184, 2.3.1, some more references should be added to back up your results and the discussion should be extended rather than just describing the results. Same with the 2.3.3, 2.5 and 2.6.
4. Page 10 line 307, 3.2, the preparation of the zein hydrolysates could be a bit confusing, re-describe the preparation processes or separate the sentence may be helpful to the readability.
Author Response
Dear reviewer,
Thank you for your patient review. Please find my detailed response in the attached file.
Kind regards,
Han

Reviewer 2 Report
In this paper, it was found that the hydrophobicity of zein was improved by mild hydrothermal-alkaline treatment. In addition, the hydrolysate obtained under the optimum hydrolysis process has better emulsifying property. Changes in protein structure lead to changes in its functional properties. The author's explanation in the section of results and discussion was also adequate. In short, the research in this paper is helpful to increase the value and utilization rate of corn by-products. However, there are a few corrections need to be done:
In Section 3.4, in line 332, the '2' in 'm2/g' should be the superscript. In line 339, and the '10' in ' A10 ' should be the subscript.
In Section 2.1, line 88-89, the grammar of this sentence of “On the contrary, Mäkinen et al. [19] disulfide-driven aggregations were not observed with extensive heating (100 ℃) at lower pH (10.5)” is wrong.
In Section 2.1, line 85, the sentence of “the bands of treated samples presented between Mw of 3.3 and 6.5 kDa” should be explained more detail or removed.
Many discussions are inappropriate, such as “2.1. Tris-tricine SDS-PAGE of zein and hydrolysates ”. Alkali hydrolysis at different temperatures was used to prepare peptides, but the molecular weight distribution was discussed by citing the article of thermal aggregation. It is suggested to discuss it again.
“2.2.3. Particle size distribution and Turbiscan stability index” I have not found Turbiscan stability index (TSI) values or curves, only BSL profiles in this section. TSI should be added.
The particle size distribution of emulsions stabilized by ZH0.6-70 after adding different concentrations of NaCl is not a single peak or normal distribution but a double peak. It is not appropriate for the author to characterize the particle size by D[3,2]. Thus, the author got the inaccurate conclusion “As shown in Table S2 and Figure 3a, the particle sizes of the emulsion droplets did not change with the increase in NaCl concentration below 200 mmol/L.” In fact, the second peak has appeared in ZH-100 mM, which may be caused by coalesced oil droplets or others.
According to the author, what causes the difference of amino acid content? As far as I know, alkali hydrolysis should only lead to the breaking of peptide bonds, will it cause changes in the molecular structure of amino acids? Is there any reference support?
Author Response
Dear reviewer,
Thank you for your professional and careful review. Please find my detailed response in the attached file.
Kind regards,
Han

Round 2
Reviewer 2 Report
The abbreviation TSI should be defined.
Author Response
Dear reviewer,
Thank you for your suggestions, and the definition of TSI has been added in the manuscript as "The Turbiscan Stability Index (TSI) is a specific parameter developed by Turbiscan Co. to compare and characterize the physical stability of various formulations with a single, comparable, and replicable number."
Please see the details in the attached file.
